# Prospective comparative analysis of noninvasive body temperature monitoring using zero heat flux technology (SpotOn sensor) compared with esophageal temperature monitoring during pediatric surgery

**Bo-Hyun Sang**⊙*, **Changjin Lee, Da Yeong Lee**

Department of Anesthesiology and Pain Medicine, CHA Bundang Medical Cener, CHA University School of Medicine, Seongnam, Republic of Korea

* bhs125@chamc.co.kr

**Data Availability Statement:** All relevant data are within the paper.

## Abstract

Maintaining body temperature in pediatric patients is critical, but it is often difficult to use currently accepted core temperature measurement methods. Several studies have validated the use of the SpotOn sensor for measuring core temperature in adults, but studies on pediatric patients are still lacking. The aim of this study was to investigate the accuracy of the SpotOn sensor compared with that of esophageal temperature measurement in pediatric patients intraoperatively. Children aged 1–8 years with American Society of Anesthesiology Physical Condition Classification I or II scheduled to undergo elective ear surgery for at least 30 min under general anesthesia were enrolled. Body core temperature was measured every 15 min after induction till the end of anesthesia with an esophageal probe, axillary probe, and SpotOn sensor. We included 49 patients, providing a total 466 paired measurements. Analysis of Pearson rank correlation between SpotOn and esophageal pairs showed a correlation coefficient (r) of 0.93 (95% confidence interval [CI] 0.92–0.94). Analysis of Pearson rank correlation between esophageal and axillary pairs gave a correlation coefficient (r) of 0.89 (95% CI 0.87–0.91). Between the SpotOn and esophageal groups, Bland-Altman analysis revealed a bias (SD, 95% limits of agreement) of -0.07 (0.17 [-0.41–0.28]). Between the esophageal and axillary groups, Bland-Altman analysis showed a bias (SD, 95% limits of agreement) of 0.45 (0.22 [0–0.89]). In pediatric patients during surgery, the SpotOn sensor showed high correlation and agreement with the esophageal probe, which is a representative core temperature measurement method.

**Funding:** The authors received no specific funding for this work.

**Competing interests:** The authors have declared that no competing interests exist.

**Abbreviations:** CI, confidence interval; SD, standard deviation; ZHF, zero-heat-flux.

## Introduction

Maintaining body temperature during perioperative period is essential, especially in pediatric patients who are susceptible to changes in body temperature. Unlike adults, pediatric patients have a larger surface area and more skin covering central structures, such as the head and torso; thus, heat loss due to conduction and radiation is more likely to occur. Therefore, pediatric patients are at a higher risk of hypothermia in the intraoperative period.

Intraoperative hypothermia is known to be associated with complications [1–3] such as delayed recovery from anesthesia, higher incidence of surgical site infections, impaired coagulation, increased blood loss, and increased cardiac morbidity. Therefore, maintaining body temperature during surgery is closely related to postoperative outcomes. It is essential to try to manage body temperature during surgery, and the most important step for body temperature management is accurate body temperature measurement.

The golden standard for core temperature monitoring is a pulmonary artery catheter. However, this method is limited, especially in pediatric patients, because of its invasive nature [4]. The area most frequently used for measuring core temperature during surgery is the distal esophagus or nasopharynx, but both are somewhat invasive and require general anesthesia. Since the axillary temperature probe is less invasive, it is often selected to measure the intraoperative core temperature in pediatric patients. However, it has disadvantages that the arm must be placed on the side of the patient so that there is no gap between the armpit and the probe, and its accuracy varies depending on the technique [5, 6]. Infrared skin temperature measurement of the upper temporal artery [7] and skin temperature measurement over the carotid artery [8, 9] have been suggested as alternatives to measuring core body temperature with less invasive methods. However, these methods seem to have limitations in their clinical use due to insufficient accuracy of core temperature measurement.

To compensate for the above shortcomings, an alternative thermometer technology called zero-heat-flux (ZHF) was developed in the 1970s [10]. The SpotOn sensor (3M Bair Hugger Temperature monitoring system) using the ZHF method comprises a thermal insulator adjacent to the skin. Once connected to the Bair Hugger control unit, the flex circuit actively regulates its temperature, creating complete insulation, which eliminates heat loss to the environment [6, 11, 12].

A previous study has shown a good correlation between measurements recorded using the SpotOn sensor and core temperature measured with a pulmonary artery thermistor during cardiac surgery [12]. Likewise, some studies demonstrated that the SpotOn sensor has good accuracy compared with nasopharyngeal [13] and esophageal probes [14]. However, few studies have been conducted in pediatric patients [15].

Thus, the purpose of this study was to evaluate the accuracy of the SpotOn sensor in comparison to that of the most commonly used esophageal temperature measurement technique to measure intraoperative body temperature in pediatric patients. In particular, we hypothesized that the SpotOn sensor is accurate in an intraoperative setting within 0.5 ˚C of reference values [12]. The accuracy and precision of axillary temperature measurement, a noninvasive monitoring method frequently used in pediatric patients, were also evaluated.

## Methods

The study was conducted with the approval of the Institutional Review Board (IRB approval number CHAMC 2019-04-005) and in accordance with the principles of the Declaration of Helsinki. Written consent to participate in this study was obtained from parents of potential participants.

The study population comprised children aged 1–8 years, with American Society of Anesthesiology Physical Condition Classification I or II, scheduled to undergo elective ear surgery for at least 30 min under general anesthesia. Patients with fragile forehead skin, systemic or forehead infections, contraindications to esophageal probe insertion, or surgery duration less than 30 min were excluded from this study.

## Protocol

Prior to entering the operating room, all patients were administered an intravascular injection of 0.04 mg kg$^{-1}$ of glycopyrrolate and 1 mg kg$^{-1}$ of ketamine. Standard monitoring was established, including electrocardiogram, pulse oximetry, and noninvasive blood pressure (5-min intervals). Subsequently, general anesthesia was induced and maintained with inhaled sevoflurane (1 to 1.5 MAC). After injection of fentanyl and rocuronium, mask ventilation was performed for an appropriate time, and an orotracheal tube was inserted. No warming device was used during surgery.

Body core temperature was measured every 15 min after induction till the end of anesthesia with an esophageal probe (Pediatric temperature probe, ETP1040, Ewha biomedics, Korea), axillary probe (TP401, Insung Medical Co., Ltd., Korea), and SpotOn sensor (3M™ Bair Hugger™ sensor, 36000, 3M Medical, USA). The esophageal probe insertion depth was calculated according to the following formula: height/5 + 5 (cm) [16]. The axillary temperature probe was attached in close contact to the patient's armpit and the SpotOn sensor was attached to the forehead.

## Statistical analysis

Data from a previous study were used to estimate the sample size required to detect significant differences in the mean temperature [13]. Analysis was performed using G*Power 3.1.9.3 with mean difference = 0.07, standard deviation (SD) = 0.22, power (β) = 0.9, and α (two-sided) = 0.05, which indicated a sample size of 44 patients. The final number of study participants was set at 49, considering a dropout rate of 10%.

Correlations between the esophageal temperature and SpotOn sensor measurements, and esophageal temperature and axillary temperature were evaluated using Spearman's rank correlation analyses. In addition, correlations between each group were also analyzed to investigate the effect of the measurement time and range of esophageal temperature (<36.5 ˚C, 36.5–37.5 ˚C, and >37.5 ˚C). We analyzed the agreement between SpotOn sensor and esophageal temperature measurements using the Bland-Altman repeated measurement data formula to adjust for within-patient correlation [17].

Continuous variables are described as mean ± SD or median and interquartile range, and categorical variables as frequencies (%), as appropriate. Analyses were performed using R software version 3.6.2 (R Foundation for Statistical Computing, Vienna, Austria).

## Results

A total of 49 patients were successfully included in the study, corresponding to a sample size of 466 data pairs. Of the 49 patients, 35 (71.4%) were male and the mean (SD) age was 45 (16) months. Anesthesia time ranged from 50 to 260 min, with an average of 2.57 h. Patient characteristics are shown in Table 1.

SpotOn temperatures were minimally higher than esophageal temperatures (mean difference 0.07 ˚C) (P = 0.0287). However, axillary temperatures were significantly lower than esophageal temperatures (mean difference 0.45 ˚C) (P < 0.0001).

**Table 1. Patient characteristics and surgical data.**

| Variables | N = 49 |
|---|---|
| Age (months) | 45 ± 16 |
| Sex (male) | 35 (71.4%) |
| Body mass index (kg/m$^2$) | 16.2 ± 1.8 |
| Anesthesia time (min) | 154 ± 43 |
| Operation time (min) | 112 ± 43 |

Analysis of Pearson rank correlation between the SpotOn and esophageal pairs showed a correlation coefficient (r) of 0.93 (95% confidence interval [CI] 0.92–0.94). The Pearson correlation analyses over time between both methods are described in Table 2 and Fig 1. Analysis of Pearson rank correlation between esophageal and axillary pairs gave a correlation coefficient (r) of 0.89 (95% CI 0.87–0.91). Pearson correlation analyses over time between both methods are as shown in Table 2 and Fig 2. Of the 466 data pairs, the zone of esophageal temperature < 36.5˚C consists of 64 data pairs, which accounted for 13.7%. The zone of esophageal temperature 36.5–37.5˚C and >37.5˚C each accounted for 362 data pairs (77.7%) and 40 data pairs (8.6%). Table 3 shows the correlation analyses for each section divided by the esophageal temperature. As shown in Table 3, when the esophageal body temperature was above 36.5 ˚C, there was a strong correlation between SpotOn and esophageal pairs, but when it was below 36.5 ˚C, a weaker correlation was found. Similar results were also observed for esophageal and axillary pairs.

Between the SpotOn and esophageal groups, Bland-Altman analysis revealed a bias (SD, 95% limits of agreement) of 0.07 (0.17 [-0.41–0.28]), indicating good agreement between SpotOn and esophageal temperature with a mean temperature range observed from 35.9 ˚C to 38.7 ˚C (Fig 3). Between the esophageal and axillary groups, Bland-Altman analysis showed a bias (SD, 95% limits of agreement) of 0.45 (0.22 [0–0.89]) (Fig 4). Bland-Altman analysis between the SpotOn and esophageal groups showed that the limits of agreement were quite narrow. However, Bland- Altman analysis between esophageal and axillary groups showed more variability, especially when the body temperature was less than 36.5 ˚C.

**Table 2. Correlation analysis of esophageal temperature and SpotOn sensor, and esophageal temperature and axillary temperature divided by time after induction.**

| Time (min) after induction | Pearson correlation coefficient (95% confidence interval) | |
|---|---|---|
| | Esophageal and SpotOn pairs | Esophageal and axillary pairs |
| 10 | 0.74 (0.58–0.84) | 0.65 (0.45–0.79) |
| 25 | 0.80 (0.68–0.89) | 0.79 (0.66–0.88) |
| 40 | 0.82 (0.7–0.9) | 0.81 (0.69–0.89) |
| 55 | 0.87 (0.77–0.92) | 0.86 (0.76–0.92) |
| 70 | 0.89 (0.82–0.94) | 0.85 (0.74–0.91) |
| 85 | 0.92 (0.85–0.95) | 0.86 (0.75–0.92) |
| 100 | 0.92 (0.85–0.95) | 0.84 (0.72–0.91) |
| 115 | 0.91 (0.84–0.95) | 0.84 (0.71–0.91) |
| 130 | 0.92 (0.84–0.96) | 0.88 (0.75–0.94) |
| 145 | 0.94 (0.86–0.97) | 0.89 (0.77–0.95) |
| 160 | 0.94 (0.85–0.98) | 0.90 (0.74–0.96) |
| 175 | 0.98 (0.93–0.99) | 0.96 (0.86–0.99) |

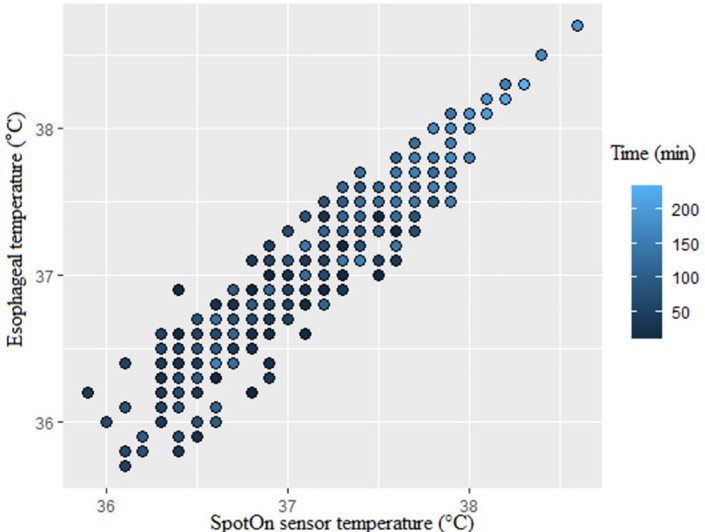

**Fig 1. Scatter plot for esophageal and SpotOn temperatures (measurement time is indicated by colored circles).**

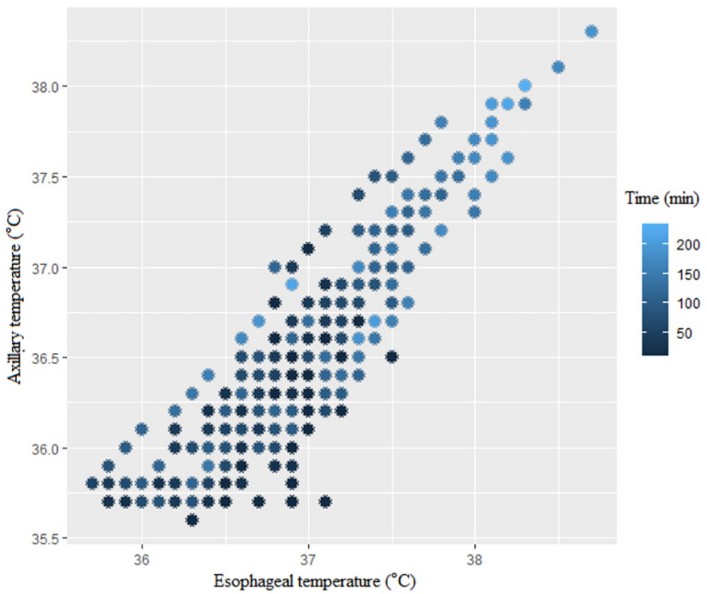

**Fig 2. Scatter plot for esophageal and axillary temperatures (measurement time is indicated by colored circles).**

**Table 3. Correlation analysis of esophageal temperature and SpotOn sensor, and esophageal temperature and axillary temperature divided by esophageal temperature.**

| | Pearson correlation coefficient (95% confidence interval) | |
|---|---|---|
| **Esophageal temperature (˚C)** | **Esophageal and SpotOn pairs** | **Esophageal and axillary pairs** |
| T<36.5 | 0.47 (0.25–0.64) | 0.34 (0.10–0.54) |
| 36.5<=T<=37.5 | 0.86 (0.83–0.88) | 0.75 (0.70–0.79) |
| T>37.5 | 0.87 (0.77–0.93) | 0.80 (0.66–0.89) |

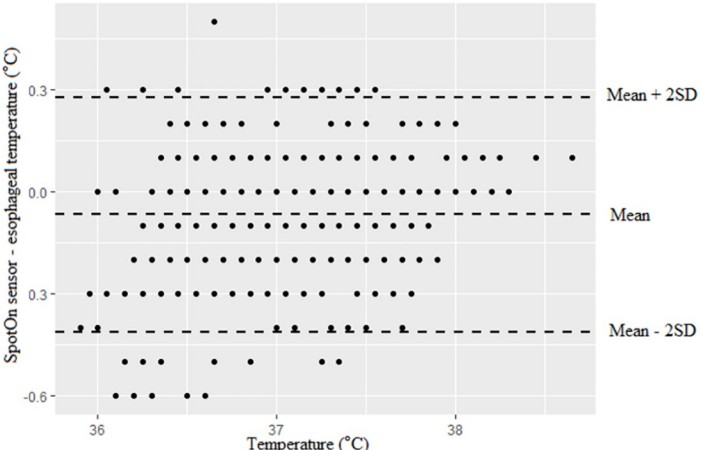

**Fig 3. Bland–Altman analysis for SpotOn sensor versus esophageal temperature monitoring.**

In the esophageal and SpotOn groups, the rate of measurement with an absolute difference within 0.5 ˚C was 98.9%. However, in the esophageal and axillary groups, the measurement rate of the absolute difference within 0.5 ˚C was only 70%.

## Discussion

The main findings of the current study can be summarized as follows. In this study, on comparison with the esophageal probe, the SpotOn sensor exhibited a bias of 0.07 in pediatric surgery. In addition, 98.9% of all measurements were within 0.5 ˚C. The correlation coefficient between the SpotOn sensor and esophageal probe was 0.93, demonstrating a good correlation. Based on these results, it can be considered that the SpotOn sensor has a tolerable accuracy for clinical use in intraoperative pediatric patients.

The results of the present study coincide with those of Carvalho et al. [15] who found a correlation of 0.86 and an average intraoperative temperature difference of 0.14 ˚C between the

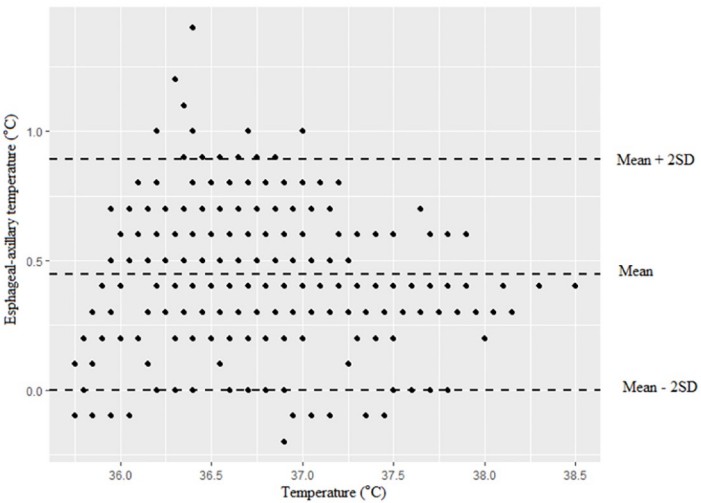

**Fig 4. Bland–Altman analysis for axillary versus esophageal temperature monitoring.**

SpotOn sensor and the esophageal thermometer. These results also show consistent similarity with adult studies showing good correlations between the SpotOn sensor and esophageal, nasopharyngeal, and sublingual thermometers or pulmonary artery catheters [12–14, 18]. The findings of the present study are consistent with those of previous pediatric [15] and adult studies [14, 18] in that the correlation and accuracy of the SpotOn sensor compared with the esophageal probe were found to be acceptable for clinical use.

Pulmonary artery, esophagus, nasopharynx, and tympanic membrane are used for core temperature monitoring. It is difficult to continuously monitor body temperature using these techniques in awake patients. Because pulmonary artery monitoring is invasive, body temperature monitoring alone is not appropriate, and its application is limited. Body temperature measurement in the nasopharynx may not be accurate while breathing through the nostrils. Furthermore, depending on the surgical site, it may not be possible to use these techniques. The SpotOn sensor can be used as an alternative core temperature monitoring method in such cases as it is noninvasive and can be used in awake patients.

Currently, esophageal temperature monitoring is used quite often to monitor core temperature during surgery. However, the use of the esophageal temperature probe is contraindicated in patients with esophageal pathology. In addition, as its use is restricted in awake patients, it cannot be used under local anesthesia and can only be used under general anesthesia. In general, the axillary probe is not recognized as a core body temperature measuring technique, but it is often used instead when other core temperature measurements are not available because of its convenience and noninvasiveness. However, the axillary probe can only measure temperature when the arm is placed on the side of the body. In addition, in this study, it was observed that the SpotOn sensor reflected the esophageal temperature more reliably than the axillary temperature in terms of correlation and accuracy. Therefore, it can be considered that the SpotOn sensor is superior to the axillary temperature probe as a noninvasive body temperature monitoring tool that can be replaced when it is impossible to measure the true core temperature. The temporal-artery thermometer [7] and skin temperature over carotid artery [8, 9], which were suggested as less invasive alternatives to measuring core temperature, were insufficiently accurate for clinical application. In comparison with these methods, the correlation and agreement between the ZHF sensor and the core temperature represented by the esophageal temperature show more reliable results.

At the beginning of the measurement, the correlation between the SpotOn sensor and the esophageal probe was low (Table 2). This may be because the ZHF technology takes time to create an isothermic tunnel; thus, the surface temperature is the same as the core temperature. In addition, when the body temperature was less than 36.5 ˚C, a weak correlation was found between the SpotOn sensor and esophageal probe (Table 3). These results suggest that the accuracy of the ZHF sensor may be limited in the initial stage after induction or when the body temperature is 36.5˚C or less. Based on these results, it is necessary to have doubts about the measured value before the ZHF system is stabilized after induction. In addition, it is recommended that caution is required in interpreting the ZHF sensor value in situations in which the body temperature can be less than 36.5˚C, that is, in the cases of emergency surgeries that tend to show unstable body temperature, or in the cases of large exposure to the surgical site. Further studies are needed to find out whether the measurement time or temperature range affects the correlation between the SpotOn sensor and esophageal probe.

Carvalho et al. mentioned that the upper limit of agreement between the esophageal probe and SpotOn sensor is 0.66 ˚C, which is higher than the cutoff value of 0.5 ˚C; thus, it might have been overestimated [15]. However, in this study, the upper limit of agreement between the esophageal probe and the SpotOn sensor was 0.28, which is within the tolerable range. At this point, it is worth noting the difference in total measurement time between the two studies.

In fact, the anesthesia time was 49 ± 19.8 min in Carvalho's study, whereas it was 154 ± 43 min in this study. This study showed a stronger correlation over time, which may have contributed to the above differences between the two studies.

This study has some limitations. First, it was not possible to define the true core temperature by using the esophageal temperature as a reference value instead of the pulmonary catheter temperature, which is the gold standard for body temperature measurement. Since the correlation between esophageal temperature and pulmonary catheter temperature has already been recognized [6], we used the esophageal temperature as a reference value. Because of the invasive nature of the pulmonary catheter, further studies should be conducted in cases where it is necessary to mount it. Second, since this study was conducted only with elective patients, it was not reviewed in emergency patients, who are vulnerable to thermoregulation. More studies are needed in emergency patients considering multiple variables involved in their management.

In conclusion, according to the results of this study, the SpotOn sensor showed high correlation and agreement with the esophageal temperature probe in intraoperative pediatric patients. Therefore, the SpotOn sensor can be considered a valuable noninvasive tool to replace the pulmonary catheter, which is the gold standard method for core temperature measurement.

## Author Contributions

**Conceptualization:** Bo-Hyun Sang.

**Data curation:** Changjin Lee, Da Yeong Lee.

**Formal analysis:** Bo-Hyun Sang.

**Investigation:** Da Yeong Lee.

**Supervision:** Bo-Hyun Sang.

**Writing – original draft:** Bo-Hyun Sang, Changjin Lee.

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
