## [Decision Letter · Decision Letter 0]

6 Jul 2022

PONE-D-22-17177Prospective comparative analysis of noninvasive body temperature monitoring using zero heat flux technology (SpotOn sensor) compared with esophageal temperature monitoring during pediatric surgeryPLOS ONE

Dear Dr. Sang,

Thank you for submitting your manuscript to PLOS ONE. After careful consideration, we feel that it has merit but does not fully meet PLOS ONE’s publication criteria as it currently stands. Therefore, we invite you to submit a revised version of the manuscript that addresses the points raised during the review process.

ACADEMIC EDITOR: The reviewers have highlighted a few items that must be addressed within the revised version of the manuscript.==============================

We look forward to receiving your revised manuscript.

Kind regards,

William M. Adams

Academic Editor

PLOS ONE

Journal Requirements:

Additional Editor Comments:

The submitted manuscript is well executed and will contribute to the body of knowledge. Please address the points outlined by the reviewers in your revision.

Reviewers' comments:

Reviewer's Responses to Questions

**Comments to the Author**

1. Is the manuscript technically sound, and do the data support the conclusions?

Reviewer #1: Partly

Reviewer #2: Yes

Reviewer #3: Yes

2. Has the statistical analysis been performed appropriately and rigorously? 

Reviewer #1: Yes

Reviewer #2: Yes

Reviewer #3: I Don't Know

3. Have the authors made all data underlying the findings in their manuscript fully available?

Reviewer #1: Yes

Reviewer #2: Yes

Reviewer #3: Yes

4. Is the manuscript presented in an intelligible fashion and written in standard English?

Reviewer #1: Yes

Reviewer #2: Yes

Reviewer #3: Yes

5. Review Comments to the Author

Reviewer #1: I would like to thank the editor for the opportunity to review the manuscript entitled "Prospective comparative analysis of noninvasive body temperature monitoring using zero heat flux technology (SpotOn sensor) compared with esophageal temperature monitoring during pediatric surgery". The paper is well-written and the study has been well designed and described.

Monitoring body temperature in the clinical setting is of great importance and one of the greatest challenge is represented by the need to measure or estimate core body temperature values with as less invasive as possible techniques due to several limitation that can depend on the setting or patient. This study provides a comparison between a non-invasive device (SpotOn) based on the ZHF technology, and a standard-use technology in ICU or surgery, as the esophageal temperature, in a specific population, i.e., pediatric surgery patients.

The results suggest a good agreement between the techniques, but some limitations can be also considered as the time needed to reach good agreement values, and some poor readings when temperature is < 36.5

Based on these considerations, the manuscript is worth of publication; however, I have few suggestions I would like to make before acceptance.

In general, I suggest to expand the references a little bit, by adding some papers and reviews on this topic.

- Introduction:

Correctly the authors state that few studies are present about the validation of ZHF in pediatric patients (lines 73-74). Can the authors cite here some of the papers that are already present in the literature, or some reviews? (e.g., Lee et al., 2021; Nemeth et al., 2020, Carvalho et al., 2019; Morettini et al., 2020; Conway et al., 2021).

- Results:

Results are well presented, although since the authors present the "temperature zones" analysis, it might be interesting to know how many readings were found in each zone for the esophageal probe. Indeed, since it is reported the lower agreements are present when temperature is < 36.5 °C, it might be important to understand how much common such zone can be during such surgery in pediatric patients.

- Discussion:

Based on these findings, I would suggest to well highlight the limitations about the validity of the ZHF system (i.e., time after induction + low body temperature), and maybe provide some sort of recommendations for its use (e.g., to wait n minutes before considering the measurement correct, or to be careful when <36.5 °C is expected).

Reviewer #2: In the present study, the authors sought to validate the non-invasive “SpotOn” measurement technic for estimating core temperature against esophageal temperature in children undergoing surgery. The authors found exceptionally good agreement of the SpotOn method. My only comment is that I wish the authors had also compared to other acceptable non-invasive methods for measuring core temperature, such as surface temperature over the carotid (Jay et al, 2013, Pediatric Anesthesia; Imani et al, 2016, Anesthesiology and pain medicine). The authors should mention this technic in their intro and discussion, however, SpotOn had better agreement in this study than surface carotid temperature in those other studies, and therefore mentioning these studies should only serve to strengthen the argument for this new technic. Minor comments below. Great job!

P3, L28 – Missing space after period

P3, L30 – Missing space after period

P3, L37 – This should be -0.07, shouldn’t it?

P6, L108 – Double ref 10

Reviewer #3: The current manuscript compares the SpotOn sensor to esophageal and axillary temperatures during elective pediatric surgery. Collectively, I have little issues with the manuscript and the statistical analysis done. In my opinion, the purpose of the study has been justified and well executed. However, the main concern I have with the findings is the weaker relationship between esophageal temperature and the SpotOn sensor at lower internal temperatures (<36.5C). In my opinion, this should be more thoroughly highlighted in the discussion, especially when the authors allude to emergency patients (whom still need to be investigated) having greater vulnerability to impaired thermoregulatory control.

6. PLOS authors have the option to publish the peer review history of their article (what does this mean?). If published, this will include your full peer review and any attached files.

Reviewer #1: No

Reviewer #2: No

Reviewer #3: No

---

## [Author Response · Author response to Decision Letter 0]

22 Jul 2022

EDITOR’S COMMENTS: 

Journal Requirements:

Thank you for your comment. We checked that our manuscript meets PLOS ONE’s style requirements and revised our manuscript according to PLOS ONE’s style requirements.

Thank you for your comment. We checked and revised our references.

Additional Editor Comments:

The submitted manuscript is well executed and will contribute to the body of knowledge. Please address the points outlined by the reviewers in your revision.

Thank you very much for your compliment. Below, we respond to the reviewers’ remarks (in bold) and include direct quotes from the manuscript (in italics) point-by-point.

REVIEWERS’ COMMENTS: 

Reviewer #1 

I would like to thank the editor for the opportunity to review the manuscript entitled "Prospective comparative analysis of noninvasive body temperature monitoring using zero heat flux technology (SpotOn sensor) compared with esophageal temperature monitoring during pediatric surgery". The paper is well-written and the study has been well designed and described.

Monitoring body temperature in the clinical setting is of great importance and one of the greatest challenge is represented by the need to measure or estimate core body temperature values with as less invasive as possible techniques due to several limitation that can depend on the setting or patient. This study provides a comparison between a non-invasive device (SpotOn) based on the ZHF technology, and a standard-use technology in ICU or surgery, as the esophageal temperature, in a specific population, i.e., pediatric surgery patients.

The results suggest a good agreement between the techniques, but some limitations can be also considered as the time needed to reach good agreement values, and some poor readings when temperature is < 36.5

Based on these considerations, the manuscript is worth of publication; however, I have few suggestions I would like to make before acceptance.

In general, I suggest to expand the references a little bit, by adding some papers and reviews on this topic.

- Introduction:

Correctly the authors state that few studies are present about the validation of ZHF in pediatric patients (lines 73-74). Can the authors cite here some of the papers that are already present in the literature, or some reviews? (e.g., Lee et al., 2021; Nemeth et al., 2020, Carvalho et al., 2019; Morettini et al., 2020; Conway et al., 2021).

 Thank you for your comment and we are sorry for not providing some references.

We cited a reference as follows.

Introduction section:

“However, few studies have been conducted in pediatric patients[15].”

Reference section:

“15. Carvalho H, Najafi N, Poelaert J. Intra-operative temperature monitoring with cutaneous zero-heat- flux-thermometry in comparison with oesophageal temperature: A prospective study in the paediatric population. Paediatr Anaesth. 2019;29(8):865-71. Epub 2019/04/30. doi: 10.1111/pan.13653. PubMed PMID: 31034706.”

- Results:

Results are well presented, although since the authors present the "temperature zones" analysis, it might be interesting to know how many readings were found in each zone for the esophageal probe. Indeed, since it is reported the lower agreements are present when temperature is < 36.5 °C, it might be important to understand how much common such zone can be during such surgery in pediatric patients.

Thank you for your advice. We totally agree to your comment. According to your comment, we specify percentage of each zone for the esophageal probe. Of the 466 data pairs, the zone of esophageal temperature < 36.5°C consists of 64 data pairs, which accounted for 13.7%. The zone of esophageal temperature 36.5-37.5°C and >37.5°C each accounted for 362 data pairs (77.7%) and 40 data pairs (8.6%). In this study, since the patients were relatively healthy and the exposure of the surgical field was very narrow, the body temperature during surgery tends to rise rather than drop. It is expected that the proportion of the low body temperature section will be higher in the case of vulnerable children in emergency surgery or in the case of wide exposure to the surgical field. We revised the manuscript as follows.

Result section: “Of the 466 data pairs, the zone of esophageal temperature < 36.5°C consists of 64 data pairs, which accounted for 13.7%. The zone of esophageal temperature 36.5-37.5°C and >37.5°C each accounted for 362 data pairs (77.7%) and 40 data pairs (8.6%).”

- Discussion:

Based on these findings, I would suggest to well highlight the limitations about the validity of the ZHF system (i.e., time after induction + low body temperature), and maybe provide some sort of recommendations for its use (e.g., to wait n minutes before considering the measurement correct, or to be careful when <36.5 °C is expected).

Thank you for your suggestion. We strongly agree to your advice. We revised the manuscript as follows.

Discussion section: “These results suggest that the accuracy of the ZHF sensor may be limited in the initial stage after induction or when the body temperature is 36.5°C or less. Based on these results, it is necessary to have doubts about the measured value before the ZHF system is stabilized after induction. In addition, it is recommended that caution is required in interpreting the ZHF sensor value in situations in which the body temperature can be less than 36.5°C, that is, in the cases of emergency surgeries that tend to show unstable body temperature, or in the cases of large exposure to the surgical site.”

Reviewer #2

 In the present study, the authors sought to validate the non-invasive “SpotOn” measurement technic for estimating core temperature against esophageal temperature in children undergoing surgery. The authors found exceptionally good agreement of the SpotOn method. My only comment is that I wish the authors had also compared to other acceptable non-invasive methods for measuring core temperature, such as surface temperature over the carotid (Jay et al, 2013, Pediatric Anesthesia; Imani et al, 2016, Anesthesiology and pain medicine). The authors should mention this technic in their intro and discussion, however, SpotOn had better agreement in this study than surface carotid temperature in those other studies, and therefore mentioning these studies should only serve to strengthen the argument for this new technic. Minor comments below. Great job!

Thank you for your advice. According to your comment, we revised the manuscript as follows.

Introduction section: “Infrared skin temperature measurement of the upper temporal artery and skin temperature measurement over the carotid artery have been suggested as alternatives to measuring core body temperature with less invasive methods. However, these methods seem to have limitations in their clinical use due to insufficient accuracy of core temperature measurement.”

Discussion section: “The temporal-artery thermometer and skin temperature over carotid artery, which were suggested as less invasive alternatives to measuring core temperature, were insufficiently accurate for clinical application. In comparison with these methods, the correlation and aggrement between the ZHF sensor and the core temperature represented by the esophageal temperature show more reliable results.”

P3, L28 – Missing space after period

Thank you for your comment. We revised the manuscript as follows.

Abstract section: “intraoperatively. Children”

P3, L30 – Missing space after period

Thank you for your comment. We revised the manuscript as follows.

Abstract section: “enrolled. Body”

P3, L37 – This should be -0.07, shouldn’t it?

Thank you for your comment and we are sorry for our mistake. We revised the manuscript as follows.

Abstract section: “-0.07”

P6, L108 – Double ref 10

Thank you for your comment and we are sorry for our mistake. We deleted one reference.

Reviewer #3

The current manuscript compares the SpotOn sensor to esophageal and axillary temperatures during elective pediatric surgery. Collectively, I have little issues with the manuscript and the statistical analysis done. In my opinion, the purpose of the study has been justified and well executed. However, the main concern I have with the findings is the weaker relationship between esophageal temperature and the SpotOn sensor at lower internal temperatures (<36.5C). In my opinion, this should be more thoroughly highlighted in the discussion, especially when the authors allude to emergency patients (whom still need to be investigated) having greater vulnerability to impaired thermoregulatory control.

Thank you for your comment and we are sorry for insufficient description regarding the weaker relationship between esophageal temperature and the SpotOn sensor at lower body temperature (<36.5°C). We revised the manuscript as follows.

Discussion section: “These results suggest that the accuracy of the ZHF sensor may be limited in the initial stage after induction or when the body temperature is 36.5°C or less. Based on these results, it is necessary to have doubts about the measured value before the ZHF system is stabilized after induction. In addition, it is recommended that caution is required in interpreting the ZHF sensor value in situations in which the body temperature can be less than 36.5°C, that is, in the cases of emergency surgeries that tend to show unstable body temperature, or in the cases of large exposure to the surgical site.”

---

## [Editor Report · Decision Letter 1]

26 Jul 2022

Prospective comparative analysis of noninvasive body temperature monitoring using zero heat flux technology (SpotOn sensor) compared with esophageal temperature monitoring during pediatric surgery

PONE-D-22-17177R1

Dear Dr. Sang,

We’re pleased to inform you that your manuscript has been judged scientifically suitable for publication and will be formally accepted for publication once it meets all outstanding technical requirements.

Kind regards,

William M. Adams

Academic Editor

PLOS ONE
---

## [Editor Report · Acceptance letter]

29 Jul 2022

PONE-D-22-17177R1 

Prospective comparative analysis of noninvasive body temperature monitoring using zero heat flux technology (SpotOn sensor) compared with esophageal temperature monitoring during pediatric surgery 

Dear Dr. Sang:

I'm pleased to inform you that your manuscript has been deemed suitable for publication in PLOS ONE. Congratulations! Your manuscript is now with our production department. 

Kind regards, 

on behalf of

Dr. William M. Adams 

Academic Editor

PLOS ONE